# Sequencing Therapy for Optimal Response in Mirikizumab (STORM)-study: A tertiary referral center study on patients with therapy-refractory ulcerative colitis

Alica Kubesch[1☉], Raul Lande[1☉], Anna Leutgöb[1], Karima Farrag[1], Iulia Dahmer[2], Katharina Stratmann-Vollrath[1], Antje Dienethal[1], Florian Alexander Michael[1], Kathrin Sprinzl[1], Stefan Zeuzem[1], Irina Blumenstein[1*]

1 Goethe University Frankfurt, University Hospital, Medical Clinic 1, Frankfurt, Germany, 2 Goethe University Frankfurt, University Hospital, Institute of Biostatistics and Mathematical Modeling, Frankfurt, Germany

☉ These authors have contributed equally.
* blumenstein@em.uni-frankfurt.de

## Abstract

### Background

Optimized drug sequencing is an emerging area of interest in the treatment of ulcerative colitis (UC). Comparative real-world data on treatment response to mirikizumab in a cohort with exposure to multiple biologic agents, particularly tumor necrosis factor (TNF)-naïve versus TNF-treated patients, remain limited. This study evaluated the therapeutic response to mirikizumab treatment in a cohort of patients with UC who were refractory to biologic therapy.

### Methods

Consecutive patients with UC treated with mirikizumab between July 01, 2023, and May 31, 2025, at a tertiary university referral center were retrospectively analyzed. The primary endpoint was 12-week clinical remission. The secondary endpoints included clinical remission and biochemical remission between weeks 24 and 50 and between weeks 60 and 80.

### Results

This study included 52 patients. Among them, 17 (32.7%) had previous exposure to ≥3 biologic agents/small molecules. The 12-week clinical remission rate was 35 of 52 patients (67.3%). There was a significant association between the treatment duration and clinical and biochemical remission. The likelihood of achieving clinical remission was 5.583 times higher after 12 weeks of intravenous mirikizumab treatment (odds ratio [OR] = 5.583, p = 0.002). Anti-TNF pretreatment had a positive effect on

**Data availability statement:** All relevant data are within the manuscript and its Supporting Information files.

**Funding:** Alica Kubesch was funded by the Bundesministerium für Forschung, Technologie und Raumfahrt (BMFTR, Federal Ministry of Research, Technology and Space) — 01EO2102 INITIALISE Advanced Clinician Scientist Program. BMFTR (Federal Ministry of Research, Technology and Space) funded the research rotation of Alica Kubesch as a clinician scientist, but did not influence the content of the research. The funder had no role in study design, data collection and analysis, decision to publish or preparation of the manuscript.

**Competing interests:** Alica Kubesch reports consulting and lecturer fees from AbbVie, Celgene/BMS, Galapagos, Johnson & Johnson, Takeda, employment and scientific collaboration from Fraunhofer Institute for Translational Medicine and Pharmacology ITMP and Fraunhofer Cluster of Excellence Immune Mediated Diseases CIMD, research funding by Sanofi and travel grants from Johnson & Johnson, AbbVie, Takeda, Galapagos and Pfizer. Raul Lande reports consulting and lecturer fees from AbbVie, Johnson & Johnson and Takeda and travel grants from Abbvie. Karima Farrag reports consulting and lecturer fees from AbbVie, Amgen, Celgene/BMS, Falk Foundation e.V., Johnson & Johnson, MSD International and Takeda and travel grants from Johnson & Johnson, Pharmacosmos and Lilly. Kathrin Sprinzl reports consulting fees and lecturer fees from AbbVie, Chemomab, Gilead, Ipsen, MSD and project funding from Gilead; Florian Alexander Michael reports lecutre fees from Pentax Medical and Dr. Weigert GmBH; Stefan Zeuzem reports consultancy and speaker's bureau fees from Abbvie, Allergan, Bi-oMarin, Gilead, Intercept, Johnson & Johnson, MSD/Merck, Novo Nordisk, SoBi, and Theratechnologies; Irina Blumenstein reports consulting and lecturer fees from Abbvie, Amgen, Biogen, Celgene/BMS, Celltrion, Falk Foundation, Fresenius Kabi, Galapagos, Johnson & Johnson, Lilly, Pharmacosmos, Pfizer, Takeda, employment and scientitic collaboration from Fraunhofer Institute for Translational Medicine and Pharmacology

biochemical remission (OR = 3.489, p = 0.021). Janus kinase inhibitor pretreatment had a negative effect on clinical remission (OR = 0.19, p = 0.019).

## Conclusion

Mirikizumab treatment had good short- and long-term efficacy in patients with UC who previously received biologic therapy. In particular, patients with prior anti-TNF therapies had favorable biochemical remission outcomes.

## Introduction

Ulcerative colitis (UC) is a chronic inflammatory bowel disease (IBD) characterized by continuous mucosal inflammation of the colon and a relapsing–remitting course. UC significantly impairs patients' quality of life and often requires long-term immunosuppressive therapy.

Several biologic agents and small molecules have become available over the last two decades. However, some patients exhibit inadequate response or loss of efficacy over time, thereby emphasizing the need for novel therapeutic options [1–5]. Among the currently available treatment options, tumor necrosis factor alpha (TNF-α) antagonists—such as infliximab, adalimumab, and golimumab—have long been a cornerstone in the management of moderate-to-severe UC. However, up to 30%–40% of patients do not respond to induction therapy with TNF-α inhibitors, and another substantial proportion loose response during maintenance therapy [6,7].

The European Crohn's and Colitis Organization guideline does not specify a ranking order for biologic therapy. Nevertheless, in some countries (such as Belgium and the United Kingdom), anti-TNF therapies must be used as the first-line biologic therapy due to country-specific regulations [8]. In Germany, if conventional therapy (such as mesalamine) is not successful, the treating physician has the medical freedom to select any licensed advanced therapy [9]. Previous studies have shown that treatment-naïve patients have a more favorable response to biologic agents compared with those with previous exposure to anti-TNF agents [10,11]. Therefore, therapeutic sequencing is an emerging concern not only in patients with anti-TNF pretreatment but also in patients refractory to several treatment lines [12].

In particular, this issue is relevant for newer biologic agents targeting the interleukin (IL)-23 pathway. Schmitt et al. have revealed that mucosal IL-23p19 is upregulated under anti-TNF therapy. Thus, anti-TNF failure may positively affect subsequent IL-23 blockade [13]. Due to the widespread use of anti-TNF therapies as the first-line treatment, the response to mirikizumab after failed anti-TNF therapy is relevant.

IL-23 has emerged as a pivotal cytokine in the pathogenesis of UC, promoting the differentiation and maintenance of Th17 cells, which are implicated in mucosal inflammation. Mirikizumab, a humanized IgG4 monoclonal antibody targeting the p19 subunit of IL-23, has been effective in inducing and sustaining remission in patients with UC, including those with prior exposure to biologic therapies. In the LUCENT-1 (induction) and LUCENT-2 (maintenance) phase 3 trials, mirikizumab significantly

ITMP and Fraunhofer Cluster of Excellence Immune Mediated Diseases CIMD. and travel grants from Pfizer, Johnson & Johnson, Lilly and Takeda. All other authors have no conflict of interest to disclose. There are no patents, products in development or marketed products associated with this research to declare. None of the commercial sources mentioned has any direct influence on the research focus, study design, or even the creation of the manuscript. The lecturer, consulting, or speaker's bureau fees relate to other services that are not related to the current manuscript. The current study was financed without direct third-party funding.

improved clinical and endoscopic outcomes compared with placebo, with a favorable safety profile [14].

Importantly, post hoc analyses from these trials have revealed that mirikizumab remains highly effective even after failed anti-TNF therapy [15]. Long-term data from the LUCENT-3 extension study confirmed sustained efficacy and safety over a 3-year treatment period [16].

Randomized controlled trials have a strong internal validity. Nonetheless, their generalizability is limited due to strict eligibility criteria. Real-world data are essential for evaluating therapeutic effectiveness and safety in heterogeneous, unselected patient populations, which is typical in routine clinical practice. Further, patients exposed to more than two prior therapies or ustekinumab treatment were excluded from the pivotal trials. In addition, only an extremely limited number of patients with previous Janus kinase (JAK) inhibitor therapy comprising solely of tofacitinib were included (2% in the placebo group vs. 3.9% in the mirikizumab group) in the analysis [14].

The current study aimed to evaluate the real-world treatment outcomes of mirikizumab in patients with moderate-to-severe UC, with a particular focus on the impact of previous therapies, particularly anti-TNF-α and JAK inhibitor exposure and pretreatment with ≥3 treatment lines.

## Materials and methods

### Study population

All patients with confirmed UC who received induction therapy with mirikizumab, with at least three full on-label infusions, were included in the analysis. The patients were treated at the IBD outpatient clinic of Goethe University Hospital. The inclusion criteria were patients who were diagnosed with UC and aged ≥18 years. The exclusion criteria were patients without a definitive UC diagnosis, those with fewer than three IV mirikizumab infusions, and those aged <18 years. The study protocols were approved by the Medical Ethics Committee of Goethe University, University Hospital Frankfurt (approval no. 2024-1652) and written informed consent was obtained from all participants.

### Study design

This was a retrospective observational study on mirikizumab treatment in patients with UC. Consecutive patients who were treated at the IBD clinic of Goethe University Hospital between July 01, 2023, and May 31, 2025, and who met the inclusion criteria were selected for the formal analysis. The patients were followed up at three time points (week 12, between weeks 24 and 50, and between weeks 60 and 80). The data was accessed for research purposes between July 1, 2025, and July 31, 2025.

### Ethical considerations

This retrospective study was approved by the local ethics committee of Goethe University, University Hospital Frankfurt (approval no. 2024−1652). Further, it was conducted in accordance with the ethical and data protection regulations.

## Study endpoints and assessments

**Efficacy.** The primary endpoint was 12-week clinical remission. The secondary endpoints were clinical remission between weeks 24 and 50 and between weeks 60 and 80; biomarker remission at week 12, between weeks 24 and 50, and between weeks 60 and 80; and improvement in clinical and biochemical disease activity parameters at week 12, between weeks 24 and 50, and between weeks 60 and 80. The clinical disease activity was assessed by calculating clinical activity scores (Simple Clinical Colitis Activity Index [SCCAI]) [17]. Biochemical disease activity was assessed by measuring fecal calprotectin (FC) levels. Clinical/biochemical remission was assumed if the clinical/biochemical parameters for disease activity indicated remission (SCCAI ≤4/FC levels ≤250 µg/g) [18,19].

**Safety/adverse events.** All adverse events (AEs) that occurred during mirikizumab treatment were documented. The AEs of special interest were major infections, malignancies, progressive multifocal leukoencephalopathy, liver injury, injection site reactions, general reactions, and hypersensitivities. The documentation included AE severity, the possible need for concomitant treatment, or changes within the biologic treatment.

**Pharmacokinetics and laboratory testing.** Blood samples were obtained at baseline (week 0), week 12, and between weeks 24 and 50. At the respective time points, general laboratory parameters were measured. All laboratory tests were conducted for routine monitoring. The patients submitted a stool sample to assess biochemical disease activity by measuring FC levels at baseline, week 12, between weeks 24 and 50, and between weeks 60 and 80.

**Statistical analyses.** Statistical analyses were conducted using the Statistical Package for the Social Sciences software version 29.0.2.0 (International Business Machine Corporation, Endicott, NY, the USA), RStudio (Posit PBC, Boston, the USA), and R (R Foundation for Statistical Computing, Vienna, Austria, including additional package "lqmm"). P-values ≤ 0.05 indicated statistically significant differences.

Numerical data were expressed as mean ± standard deviation for normally distributed variables, and as median and interquartile range (IQR) for non-normally distributed variables. Categorical data were presented as absolute and relative frequencies.

To compare numeric variables between groups, the *t*-test was used for normally distributed data and the Mann–Whitney U test (U) for non-normally distributed data. The Shapiro–Wilk test was utilized to examine normality.

Categorical variables such as clinical/biochemical remission (defined by sex, previous biologic therapy or treatment with >1 type of biologic agent) were compared between groups using the chi-square test ($\chi^2$).

The McNemar test was employed to examine differences between the use of oral steroids at baseline and week 12.

For the analyses of the variables measured at several time points for each patient, regression models (logistic, median) with mixed effects were used. Mixed-effects models are appropriate for describing the associations between a response variable and covariates (called fixed effects) in data that are grouped according to one or more classification factors (called random effects). By including a common random effect for the data in the same group, the group properties are considered when estimating model parameters. The random factor in all statistic models was the patient identifier (ID) since the data consisted of repeated measurements per patient over time.

For the analysis of the evolution of the proportion of patients with clinical/biochemical remission over time binary logistic regression analyses with mixed effects were used. Further multiple logistic regression analyses were conducted to assess the effect of previous therapies and colonic infestation pattern on clinical/biochemical remission.

The time evolution of FC levels and SCCAI was analyzed using multiple median regressions with mixed effects, using patient as a random effect and time and medication history as fixed effects.

To address missing values in the dataset, the Little's Missing Completely at Random test was used.

## Results

### Study population

In total, 52 patients were included in this analysis. Among them, 17 (32.5%) were previously treated with ≥3 biological agents/small molecules, and 35 (67.33%) received <3 biological agents/small molecules. Further, 27 (51.9%) patients were

managed with anti-TNF, and 25 (48.1%) were anti-TNF-naïve. Seventeen (32.5%) patients had a previous history of JAK inhibitor treatment, and 35 (67.33%) were JAK inhibitor-naïve. Table 1 shows the characteristics of the patients at baseline.

## Subgroup analysis of the characteristics of the patients according to the type of pretreatment

**Anti-TNF pretreatment.** There was no statistically significant difference between the anti-TNF-treated and anti-TNF-naïve groups in terms of baseline characteristics such as age, sex, BMI (body mass index), disease duration, and FC and C-reactive protein (CRP) levels (S1 Table).

**JAK inhibitor pretreatment.** The JAK inhibitor-treated and JAK inhibitor-naïve groups did not significantly differ in terms of baseline characteristics such as age, sex, BMI, disease duration, and FC and CRP levels (S2 Table).

**Ustekinumab pretreatment.** A statistically significant difference was observed in terms of sex distribution between the ustekinumab-treated and ustekinumab-naïve group. The ustekinumab-treated group had a significantly higher proportion of female patients than the ustekinumab-naïve group. There was no statistically significant difference between the ustekinumab-treated and ustekinumab-naïve groups regarding other characteristics such as age, BMI, disease duration, and FC and CRP levels (S3 Table).

**Vedolizumab pretreatment.** There was no statistically significant difference between the vedolizumab-treated and vedolizumab-naïve groups regarding baseline characteristics such as age, sex, BMI, disease duration, and FC and CRP levels (S4 Table).

## Efficacy

**Clinical remission at week twelve.** The 12-week clinical remission rate was 35 of the 52 patients (67.3%). Nine patients (17.3%) did not achieve clinical remission at week 12. The data of 8 patients was missing. The likelihood of achieving clinical remission was 5.583 times higher after 12 weeks of intravenous mirikizumab treatment compared with baseline (odds ratio [OR] = 5.583, p = 0.002) (Table 2).

**Clinical remission between weeks 24 and 50.** The clinical remission rate between weeks 24 and 50 was 33 of the 52 patients (63.5%). Four patients (7.7%) did not achieve clinical remission between weeks 24 and 50. The data of 15 patients was missing. The likelihood of achieving clinical remission was 16.568 times higher after 24–50 weeks of intravenous mirikizumab treatment compared with baseline (OR = 5.583, p < 0.001) (Table 2).

**Clinical remission between weeks 60 and 80, interims analysis.** Thirteen patients reached the period between weeks 60 and 80. In total, 11 (84.6%) of these 13 patients exhibited clinical remission between weeks 60 and 80. Statistical analysis could not be performed at this point in time (Table 2).

**Clinical remission: influence of pretreatment.** Table 3 shows the results of the multiple logistic analysis with clinical remission as a dependent variable and pretreatment as a predictor. There was a statistically significant negative association between JAK inhibitor pretreatment and clinical remission (OR = 0.19, p = 0.019).

A statistically significant association was not observed between anti-TNF pretreatment, vedolizumab pretreatment, ustekinumab pretreatment, or ≥3 previous biologic agent/small molecule therapies and clinical remission.

**Clinical remission: Effect of colonic infestation pattern.** There was no statistically significant association between the colonic infestation pattern and clinical remission (S5 Table).

**Clinical response: Improvement in SCCAI.** Fig 1 shows the evolution of the SCCAI at the observed time points. The median SCCAI over the observed periods was lower compared with that at baseline.

A median regression analysis on the time evolution of the SCCAI was performed.

A longer treatment duration was associated with a lower SCCAI (p < 0.001) (S6 Table).

There was no statistically significant association between pretreatment and SCCAI for all tested medications and ≥3 biologic agent/small molecule therapies (S6 Table).

**Table 1. Characteristics of the patients at baseline with percentages for categorical variables, mean±standard deviation for normally distributed data, and median with interquartile range for non-normally distributed data.**

|  | Values |
|---|---|
| Age<br>n, median (IQR) | n=52, 38 (25) |
| Female sex<br>n (%) | n=52, 22 (42.3) |
| BMI<br>n, mean (SD) | n=46, 25.11 (5.75) |
| Weight<br>n, median (IQR) | n=45, 72.0 (24.5) |
| Disease duration (years)<br>n, median (IQR) | n=52, 7.0 (10) |
| Active disease (clinical)<br>n (%) | n=39, 21 (40.4) |
| Active disease (biochemical)<br>n (%) | n=44, 30 (57.7) |
| Extended induction#<br>n (%) | n=52, 27 (51.9) |
| Re-induction*<br>n (%) | n=52, 9 (17.2) |
| Treatment duration (weeks)<br>n, median (IQR) | n=52, 44.36 (33.46) |
| Oral steroids<br>n (%) | n=52, 16 (30.8) |
| Topical steroids<br>n (%) | n=52, 5 (9.6) |
| Previous therapy with biologic agents/small molecules<br>n (%) | n=52, 43 (82.7) |
| ≥3 biologic agent/small molecule therapies<br>n (%) | n=52, 17 (32.7) |
| Anti-TNF treatment<br>n (%) | n=52, 27 (51.9) |
| JAK inhibitor treatment<br>n (%) | n=52, 17 (32.7) |
| Ustekinumab treatment<br>n (%) | n=52, 16 (30.8) |
| Vedolizumab treatment<br>n (%) | n=52, 26 (50) |
| Ozanimod treatment<br>n (%) | n=52, 3 (5.8) |
| FC levels at baseline (μg/g)<br>n, median (IQR) | n=45, 644 (1917) |
| SCCAI at baseline<br>n, median (IQR) | n=42, 5.0 (5) |
| CRP level at baseline (mg/dL)<br>n, median (IQR) | n=41, 0.39 (1.4) |
| Proctitis<br>n (%) | n=52, 6 (11.5) |
| Rectosigmoiditis<br>n (%) | n=52, 6 (11.5) |

*(Continued)*

**Table 1.** (Continued)

|  | Values |
|---|---|
| Left-sided colitis n (%) | n = 52, 12 (23.1) |
| Pancolitis n (%) | n = 52, 24 (46.2) |
| Pouchitis n (%) | n = 52, 5 (9.6) |

BMI, body mass index; CRP, C-reactive protein; FC, fecal calprotectin; IQR, interquartile range; JAK, Janus kinase, SCCAI, Simple Clinical Colitis Activity Index; SD, standard deviation; TNF, Tumor necrosis factor. *Re-induction = on-label switch to intravenous route after subcutaneous therapy, #Extended induction = on induction therapy with a total of six infusions every 4 weeks instead of three in standard induction.

**Table 2.  Results of the mixed effects logistic regression analysis of time as a predictor associated with clinical remission.**

|  | Analysis | |
|---|---|---|
| Predictor | p value | OR (95% CI) |
| Time (week 12) | **0.002** | **5.538 (1.914–16.024)** |
| Time (weeks 24–50) | **<0.001** | **16.568 (3.893–70.509)** |
| Time (weeks 60–80) | Not evaluable | |

**Table 3.  Results of the multiple logistic regression analysis of the pretreatment associated with clinical remission.**

|  | Analysis | |
|---|---|---|
| Predictors | p value | OR (95% CI) |
| Anti-TNF pretreatment | 0.108 | 0.358 (0.102–1.258) |
| JAK inhibitor pretreatment | **0.019** | **0.190 (0.048–0.759)** |
| Ustekinumab pretreatment | 0.970 | 1.025 (0.272–3.866) |
| Vedolizumab pretreatment | 0.885 | 1.148 (0.265–3.151) |
| ≥3 biologic agent/small molecule therapies | 0.405 | 0.582 (0.161–2.100) |

JAK, Janus kinase, TNF, Tumor necrosis factor.

**Biochemical remission at week twelve.**  The 12-week biochemical remission rate was 13 of the 52 patients (25%). Twenty patients (38.5%) did not achieve biochemical remission at week 12. The data of 19 patients was missing. For patients who received three intravenous mirikizumab doses, there was no statistically significant increase in the biochemical remission rates at week 12 compared with baseline. There was a non-significant trend for the increase in the 12-week biochemical remission (Table 4).

**Biochemical remission between weeks 24 and 50.**  The biochemical remission rate between weeks 24 and 50 was 22 of the 52 patients (42.3%). Thirteen patients (25%) did not achieve biochemical remission between weeks 24 and 50. The data of 17 patients was missing. The likelihood of achieving biochemical remission was 4.34 times higher after 24–50 weeks of intravenous mirikizumab treatment (OR = 4.34, p = 0.006) (Table 4).

**Biochemical remission between weeks 60 and 80.**  Thirteen patients reached biochemical the period between weeks 60 and 80. The biochemical remission rate between weeks 60 and 80 was 8 of the 13 patients (61.5%). Three of

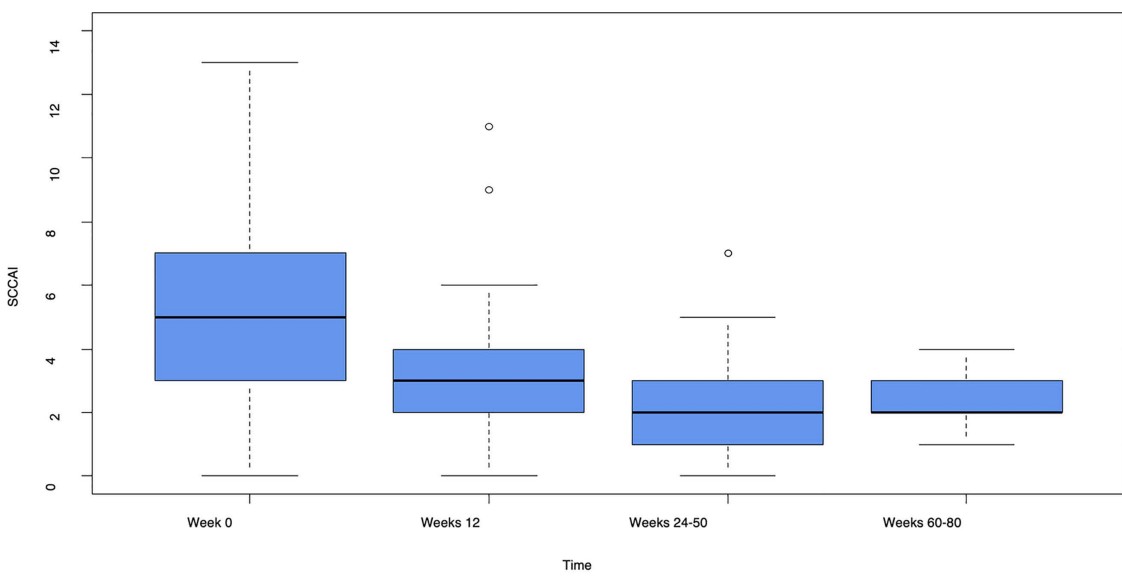

**Time evolution of SCCAI**

SCCAI, Simple Clinical Colitis Activity Index

**Fig 1. Time evolution of the Simple Clinical Colitis Activity Index.**

**Table 4. Results of the mixed effects logistic regression analysis of time as a predictor associated with biochemical remission.**

| Predictors | Analysis | |
|---|---|---|
| | p value | OR (95% CI) |
| Time (week 12) | 0.386 | 1.592 (0.552–4.587) |
| Time (weeks 24–50) | **0.006** | **4.343 (1.543–12.223)** |
| Time (weeks 60–80) | **0.048** | **5.371 (1.017–28.361)** |

these 13 patients (23.1%) did not achieve biochemical remission between weeks 60 and 80. The data of two patients was missing. The likelihood of achieving biochemical remission was 5.37 times higher after 60–80 weeks of intravenous mirikizumab treatment (OR = 5.37, p = 0.048) (Table 4).

**Biochemical remission: Influence of pretreatment.** Table 5 shows the results of the multiple logistic regression analysis of the influence of pretreatment on biochemical remission. The likelihood of achieving biochemical remission in patients with anti-TNF pretreatment was 3.489 times higher than that in anti-TNF-naïve patients (OR = 3.489, p = 0,021). There was no statistically significant association between vedolizumab pretreatment, ustekinumab pretreatment, JAK inhibitor pretreatment, or ≥3previous biologic agent/small molecule therapies and biochemical remission.

**Biochemical remission: Effect of colonic infestation pattern.** There was a statistically significant association between isolated rectal inflammation and biochemical remission (OR = 6.474, p = 0.041) (S7 Table). No statistically significant association was observed between left-sided colitis or pancolitis and biochemical remission, probably due to the inclusion of a small number of patients. Left-sided inflammation might have a non-significant advantage compared with pancolitis.

**Table 5. Results of the multiple logistic regression analysis of pretreatment as a predictor associated with biochemical remission.**

| Predictors | Analysis | |
|---|---|---|
| | p-values | OR (95% CI) |
| Anti-TNF pretreatment | **0.021** | **3.489 (1.209–10.072)** |
| JAK inhibitor pretreatment | 0.189 | 0.462 (0.145–1.471) |
| Ustekinumab pretreatment | 0.104 | 2.563 (0.821–8.003) |
| Vedolizumab pretreatment | 0.240 | 0.522 (0.176–1.552) |
| ≥3 biologic agent/small molecule therapies | 0.170 | 2.192 (0.711–6.760) |

JAK, Janus kinase, TNF, Tumor necrosis factor.

**Biochemical response: Improvement in FC levels.** Fig 2 shows the evolution of the FC levels at the observed time points. The median FC level at the observed periods was lower than that at baseline.

A longer treatment duration was associated with lower FC levels (p = 0.003 at week 12 and p < 0.001 between weeks 24 and 50 and between weeks 60 and 80) (S8 Table).

There was no statistically significant association between pretreatment and the FC levels for all tested medications and ≥3 biologic agent/small molecule therapies (S8 Table).

**Concomitant oral steroids.** In total, 16 (30.8%) patients were on oral steroids at baseline. The patients taking oral steroids at baseline had a median of 2 (IQR: 2) previous biological therapies. Among them, 8 (50%) were previously treated with anti-TNF therapy, 4 (25%) with JAK inhibitor therapy, 3 (18.8%) with ustekinumab therapy, and 10 (62.5) with vedolizumab therapy.

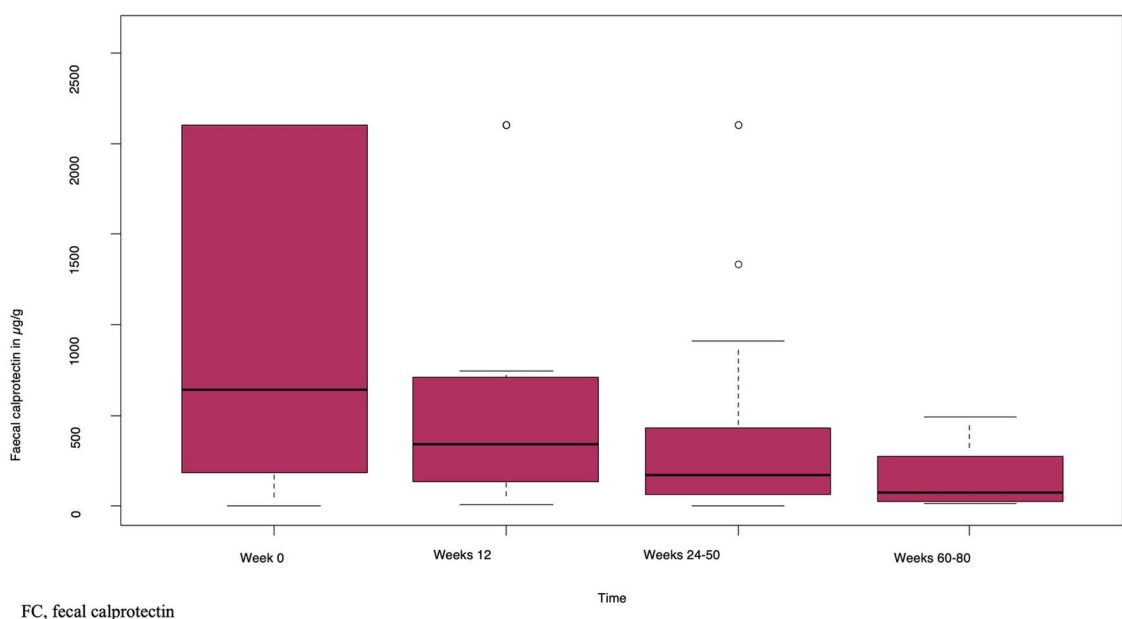

FC, fecal calprotectin

**Fig 2. Time evolution of the fecal calprotectin levels.**

Only 7 (43.8%) of 16 patients remained on steroid treatment at week 12, which indicated a significant reduction (p = 0.008). However, seven patients who were not on oral steroids at baseline were managed with oral steroids at week 12. Thus, the total number of patients who were taking oral steroids was 14. Seven patients were also managed with biologic agents, with a median of 2 (IQR: 3) previous biological therapies. Among them, 3 (42.9%) had more than two previous biological therapies, and four (57.1%) received anti-TNF therapy.

**Safety/adverse events.** After the first infusion and the first infusion of re-induction, one patient exhibited patchy reddish-livid discoloration in both legs and arms and edema in the feet. These symptoms completely regressed after one week. After switching to subcutaneous application, no similar symptoms were observed.

One patient developed pneumococcal pneumonia during the mirikizumab therapy. The condition healed completely after antibiotic therapy, and no inpatient treatment was required. A coincidental occurrence of the infection could be considered, and no further major events were documented.

**Analysis of missing data.** Based on the Little's Missing Completely at Random test, there was no evidence opposing the hypothesis that the data were missing in a completely random manner ($\chi^2$ (218) = 211.768, p = 0.606).

## Discussion

Due to the increasing number of therapeutic options for IBD, there is a need for smart therapeutic sequencing, which can provide patients with the best personalized early treatment option during the course of their disease.

Current real-world information on mirikizumab has shown encouraging outcomes, including steroid-free remission and favorable safety in both biologic-naïve and biologic-treated patients [20]. Nevertheless, comparative real-world data on treatment response in TNF-naïve versus TNF-treated patients remain limited.

To the best of our knowledge, the current study first provided real-world data on anti-IL-23 treatment with mirikizumab in a UC cohort who had extensive therapeutic experience, with more than one-third of them having been exposed to ≥3 advanced treatment lines. In this study, 32.7% of the patients were pretreated with ≥3 biologic agents/small molecules and 30.8% with ustekinumab prior to mirikizumab therapy. Patients who met the abovementioned criteria would have been excluded from the LUCENT trial [14].

Based on data of the current study, mirikizumab has extremely good clinical and biochemical efficacy in this cohort over a period of >1 year. Further, anti-TNF pretreatment may have a favorable influence on biochemical remission over time.

Twelve-week clinical remission, which was the primary endpoint, was achieved in 35 (67.3%) of the 52 patients. The 12-week clinical remission rate in this cohort was higher compared with that in the LUCENT trial (24.2%). This finding might be caused by the higher remission rates at baseline in the current study. This study included 21 (42%) patients with clinical remission (SCCAI < 5) at baseline. These patients required a treatment switch despite being in clinical remission due to factors such as intolerance and the need for off-label interval shortening of previous therapies. Only patients with clinical activity and endoscopic inflammation were included in the LUCENT trial [14]. Takagi et al. reported that the 12-week clinical remission rate in a Japanese real-world cohort with UC who received mirikizumab treatment was 44.2% (23/52). In their study, only two (3.8%) patients who were in clinical remission at baseline were included [21].

Considering the effect of pretreatment has a high clinical relevance. Therefore, the association between different biological pretreatments and the clinical/biochemical remission rates was explored. Results showed a statistically significant positive association between anti-TNF pretreatment and biochemical remission (odds ratio = 3.49, p = 0.021). In a subsequent LUCENT trial subgroup analysis conducted by Hart et al., a response rate of 78.8% was still observed even after failed anti-TNF therapy, at least in terms of clinical response at week 12 or 24 [15]. These findings are in accordance with those of a study on patients with Crohn's disease, which showed that the expansion of apoptosis-resistant intestinal TNFR2+IL23R+T cells is associated with a lack of response to anti-TNF therapy [13]. An optimal treatment sequence or combination therapy of anti-TNF and anti-IL-23 drugs could be based on this theoretical foundation.

Singh et al. performed a network meta-analysis exploring the therapeutic response to ustekinumab and tofacitinib after anti-TNF exposure. Their study showed that the drugs had good efficacy after failed infliximab treatment in UC [22]. Both drugs were superior to vedolizumab or a second anti-TNF drug (adalimumab) [22]. Considering these results, anti-IL-23 therapy with mirikizumab may also be considered efficacious after failed anti-TNF therapy.

Interestingly, the current analysis showed a statistically significant negative association between JAK inhibitor pretreatment and clinical remission (odds ratio = 0.19, p = 0.019). This effect could be explained by the disease severity. The group of patients pretreated with JAK inhibitors (n = 17) had a median of three previous therapies (IQR: 2) and a median FC level of 1421 µg/g (IQR: 1901 µg/g) at baseline. Further, only 3 (17.6%) of the 17 patients were in clinical remission at baseline. In contrast, the median FC level of JAK inhibitor-naïve patients was 525 µg/g (IQR: 1938 µg/g), and 15 (42.9%) of 35 patients were in clinical remission at baseline.

There was no statistically significant association between anti-TNF, vedolizumab, or ustekinumab pretreatment and clinical remission with mirikizumab treatment at any timepoint during the treatment. These findings are consistent with those of Takagi et al.. In their cohort, similar clinical remission rates were also observed at week 12 between patients who had previously been treated with anti-TNF or ustekinumab [21]. Compared with the results obtained from Japan and the current findings, the patients pretreated with anti-TNF in the LUCENT trial had a lower clinical remission rate at week 12 (15.7% vs. 29.3%). However, statistical significance was not reached [14]. In the current cohort, the effect of anti-TNF or ustekinumab pretreatment was not significant, and this finding may also be attributed to the small number of cases.

This study first provided long-term real-world data as the follow-up extended up to 80 weeks. A significant increase in biochemical remission was observed between weeks 60–80 (OR = 5.37, p = 0.048). In total, 11 (84.6%) of 13 patients achieved clinical remission between weeks 60–80. Two patients had missing data. Based on these results, mirikizumab is still effective even in a pretreated cohort after more than 1 year of therapy.

Concomitant steroid use is common in patients with active disease and multiple biological pretreatments. In the current cohort, 16 (30.8%) of 52 patients were taking systemic steroids at baseline. This exceeds the proportion of patients treated with systemic steroids at baseline in the study of Takagi et al. The lower rate of systemic steroid use in the Japanese cohort may be explained by the high intake of calcineurin inhibitors. In their cohort, 16 (30.8%) of 52 patients were receiving concomitant tacrolimus, which is widely used in Japan [21]. In the current study, none of the patients received calcineurin-inhibitor therapy. Further, 56.3% of the patients who were on systemic steroids at baseline achieved complete withdrawal of corticosteroid therapy by week 12. Within 12 weeks, seven patients required systemic corticosteroid. Therefore, it remains debatable whether initiating corticosteroid treatment earlier—unless unavoidable—might facilitate an earlier discontinuation. After 50 weeks, all patients could be weaned off from steroid therapy.

The use of on-label extended induction opens a novel avenue for optimized treatment response in patients treated with mirikizumab. In the current study, induction was extended in 27 (51.9%) of 52 patients. This percentage is slightly higher than that in the study of Takagi et al. (18 [38.3%] of 47) [21]. The high proportion of extended induction in the current cohort could be explained by the higher proportion of patients with historically multiple biological previous therapies. In these patients, therapeutic response is often delayed, and the success rates are reduced.

The limitations of the current study include the absence of endoscopic association and patient reported come, limited long-term follow-up data beyond 1 year, and its single-center, retrospective design. The lack of an endoscopic endpoint is a particular point of criticism. Due to the retrospective nature of the study and the resulting lack of clinical necessity for a follow-up endoscopy in patients whose symptoms had improved, no endoscopic data could be collected. The significance of the results would certainly be higher if endoscopic endpoints of higher quality had been used.

Due to the study design, some data on the respective time points were missing, which has been taken into account in the analysis of missing data. Results showed that the data were missing in a random manner. Isolated proctitis was significantly associated with biochemical remission (OR = 6.47, p = 0.041). However, due to the small number of cases (n = 6), this result should be interpreted with caution. Nevertheless, further studies with a larger number of cases should be performed to validate this possible association.

## Conclusions

In conclusion, even in a therapy-refractory UC cohort who received multiple pretreatments, mirikizumab is an effective and low-risk therapeutic option. The current results indicate that patients with anti-TNF pretreatment could benefit from mirikizumab therapy. Moreover, a previous JAK inhibitor therapy might be associated with an unfavorable response to mirikizumab therapy.

## Supporting information

**S1 Table. Characteristics of the anti-TNF-treated and anti-TNF-naïve patients at baseline** with percentages for categorical variables, mean ± standard deviation for normally distributed data, and median with interquartile range for non-normally distributed data.
(PDF)

**S2 Table. Characteristics of the JAK inhibitor-treated and JAK inhibitor-naïve patients at baseline** with percentages for categorical variables, mean ± standard deviation for normally distributed data, and median with interquartile range for non-normally distributed data.
(PDF)

**S3 Table. Characteristics of the ustekinumab-treated and ustekinumab-naïve patients at baseline** with percentages for categorical variables, mean and standard deviation for normally distributed data, and median with interquartile range for non-normally distributed data.
(PDF)

**S4 Table. Characteristics of the vedolizumab-treated and vedolizumab-naïve patients at baseline** with percentages for categorical variables, mean ± standard deviation for normally distributed data, and median with interquartile range for non-normally distributed data.
(PDF)

**S5 Table. Results of the multiple logistic regression analysis of colonic infestation pattern associated with clinical remission.**
(PDF)

**S6 Table. Results of the multiple median regression analysis with mixed effects for SCCAI as a dependent variable.**
(PDF)

**S7 Table. Results of the multiple logistic regression analysis of colonic infestation pattern associated with biochemical remission.**
(PDF)

**S8 Table. Results of the multiple median regression analysis with mixed effects for FC as a dependent variable.**
(PDF)

## Author contributions

**Conceptualization:** Alica Kubesch, Raul Lande, Irina Blumenstein.

**Data curation:** Alica Kubesch, Raul Lande, Anna Leutgöb, Iulia Dahmer.

**Formal analysis:** Alica Kubesch, Raul Lande, Anna Leutgöb, Iulia Dahmer.

**Funding acquisition:** Stefan Zeuzem.

**Investigation:** Alica Kubesch, Raul Lande, Anna Leutgöb, Karima Farrag, Katharina Stratmann-Vollrath, Antje Dienethal, Florian Alexander Michael, Kathrin Sprinzl, Irina Blumenstein.

**Methodology:** Alica Kubesch, Raul Lande, Irina Blumenstein.

**Project administration:** Irina Blumenstein.

**Resources:** Alica Kubesch, Raul Lande, Stefan Zeuzem, Irina Blumenstein.

**Supervision:** Irina Blumenstein.

**Validation:** Alica Kubesch, Raul Lande, Iulia Dahmer, Irina Blumenstein.

**Visualization:** Alica Kubesch, Raul Lande, Irina Blumenstein.

**Writing – original draft:** Alica Kubesch, Raul Lande, Irina Blumenstein.

**Writing – review & editing:** Alica Kubesch, Raul Lande, Karima Farrag, Katharina Stratmann-Vollrath, Antje Dienethal, Florian Alexander Michael, Kathrin Sprinzl, Stefan Zeuzem, Irina Blumenstein.

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
