## [Decision Letter · Decision Letter 0]

25 Sep 2025

Dear Dr.  Blumenstein, 

Thank you for submitting your manuscript to PLOS ONE. After careful consideration, we feel that it has merit but does not fully meet PLOS ONE’s publication criteria as it currently stands. Therefore, we invite you to submit a revised version of the manuscript that addresses the points raised during the review process before considering the manuscript further.

If applicable, we recommend that you deposit your laboratory protocols in protocols.io to enhance the reproducibility of your results. Protocols.io assigns your protocol its own identifier (DOI) so that it can be cited independently in the future. For instructions, see: https://journals.plos.org/plosone/s/submission-guidelines#loc-laboratory-protocols . Additionally, PLOS ONE offers an option for publishing peer-reviewed Lab Protocol articles, which describe protocols hosted on protocols.io. Read more information on sharing protocols at https://plos.org/protocols?utm_medium=editorial-email&utm_source=authorletters&utm_campaign=protocols .

We look forward to receiving your revised manuscript.

Kind regards,

Vipula Rasanga Bataduwaarachchi, MD

Academic Editor

PLOS ONE

Journal Requirements:

When submitting your revision, we need you to address these additional requirements

“Alica Kubesch was funded by the Bundesministerium  für Forschung, Technologie und Raumfahrt (BMFTR, Federal Ministry of Research, Technology and Space) ‒ 01EO2102 INITIALISE Advanced Clinician Scientist Program.”

“A.K. reports consulting and lecturer fees from AbbVie, Celgene/BMS, Galapagos, Johnson & Johnson and Takeda, R.L. reports consulting and lecturer fees from AbbVie, Johnson & Johnson and Takeda;  K.F. reports consulting and lecturer fees from AbbVie, Amgen, Celgene/BMS, Falk Foundation e.V., Johnson & Johnson, MSD International and Takeda; K.S. (Kathrin Sprinzl) reports consulting fees, lecturer fees and project funding from AbbVie, Chemomab, Gilead, Ipsen, MSD; FAM reports lecutre fees from Pentax Medical and Dr. Weigert GmBH; S.Z. reports consultancy and speaker’s bureau fees from Abbvie, Allergan, Bi-oMarin, Gilead, Intercept, Johnson & Johnson, MSD/Merck, Novo Nordisk, SoBi, and Theratechnologies; I.B. reports consulting and lecturer fees from Abbvie, Amgen, Biogen, Celgene/BMS, Celltrion, Falk Foundation, Fresenius Kabi, Galapagos, Johnson & Johnson, Lilly, Pharmacosmos, Pfizer and Takeda

All other authors have no conflict of interest to disclose.

We note that you received funding from a commercial source: “Abbvie, Amgen, Biogen, Celgene/BMS, Celltrion, Falk Foundation, Fresenius Kabi, Galapagos, Johnson & Johnson, Lilly, Pharmacosmos, Pfizer and Takeda”

5. In the online submission form, you indicated that “The data presented in this study are available on request from the corresponding author. The data are not publicly available due to data protection and ethical regulations.”

7. Your ethics statement should only appear in the Methods section of your manuscript. If your ethics statement is written in any section besides the Methods, please delete it from any other section.

8. Please include captions for your Supporting Information files at the end of your manuscript, and update any in-text citations to match accordingly. Please see our Supporting Information guidelines for more information: http://journals.plos.org/plosone/s/supporting-information

Reviewer's Responses to Questions

**Comments to the Author**

1. Is the manuscript technically sound, and do the data support the conclusions?

Reviewer #1: No

Reviewer #2: Yes

2. Has the statistical analysis been performed appropriately and rigorously?

Reviewer #1: Yes

Reviewer #2: Yes

3. Have the authors made all data underlying the findings in their manuscript fully available?

Reviewer #1: Yes

Reviewer #2: Yes

4. Is the manuscript presented in an intelligible fashion and written in standard English?

Reviewer #1: Yes

Reviewer #2: Yes

Reviewer #1: While the retrospective study is acknowledged, the inclusion criterion of “≥3 infusions” requires further clarification.

Clinical remission was indicated by SCCAI ≤4 and biochemical remission by FC ≤250 µg/g. No endoscopic remission data was included. Even in retrospective settings, lack of endoscopic correlation limits clinical value; this should be addressed in the debate.

The statistical models (mixed effects logistic regression, median regression) are appropriate, however the paper might benefit from more justification and how patient-level random effects were included.

Reviewer #2: Greetings

The subiect of this manuscript is important, this study provided real-world data on anti-IL-23 treatment

with mirikizumab in a UC cohort who had extensive therapeutic experience, with more than

one-third of them having been exposed to ≥3 advanced treatment lines. But dear authors kindly note the minor points and edit them.

Minor points,

1. In this manuscript, the pronoun "Our" (15 times) was used. In scientific writing, it is better to avoid the pronouns. Please replace them with formal scientific expressions such as "This study," "The present study," or "The current study."

2.Several references [ 5,6,17,18 and 19] are old. Please try to cite more recent references.

3- Kindly I suggest to add conclusion section after the discussion section.

Kind regards

**Do you want your identity to be public for this peer review?** For information about this choice, including consent withdrawal, please see our Privacy Policy

Reviewer #1: No

Reviewer #2: No

---

## [Author Response · Author response to Decision Letter 1]

1 Oct 2025

Dear Reviewers,

we thank you for the constructive and important critiques, which helped to improve our revised manuscript significantly. We have reworked the manuscript. The changes are highlighted yellow and we have addressed your comments point by point below.

Reviewer #1 remarks

1. While the retrospective study is acknowledged, the inclusion criterion of “≥3 infusions” requires further clarification.

This indeed seems confusing; we have have adjusted the statement (line 105).

2. Clinical remission was indicated by SCCAI ≤4 and biochemical remission by FC ≤250 µg/g. No endoscopic remission data was included. Even in retrospective settings, lack of endoscopic correlation limits clinical value; this should be addressed in the debate.

Thank you very for this important comment. The issue of the missing endoscopic endpoint has now been emphasized more strongly in the discussion (lines 452-456)

3. The statistical models (mixed effects logistic regression, median regression) are appropriate, however the paper might benefit from more justification and how patient-level random effects were included.

We understand the confusion surrounding the application of the statistical model. We have therefore added a further explanation in the Methods section (lines 170-184).

Reviewer #2 remarks

1. In this manuscript, the pronoun "Our" (15 times) was used. In scientific writing, it is better to avoid the pronouns. Please replace them with formal scientific expressions such as "This study," "The present study," or "The current study."

We agree and have revised the whole manuscript regarding the pronoun “our” and we have replaced it by formal expressions.

2. Several references [ 5,6,17,18 and 19] are old. Please try to cite more recent references.

In fact, we have replaced all sources you mentioned with more recent ones, with the exception of source 17. This is the original work, including validation of the SCCAI score.

3. Kindly I suggest to add conclusion section after the discussion section.

We absolutely agree. The conclusion section has been added (line 465).

we thank you for the constructive and important critiques, which helped to improve our revised manuscript significantly. We have reworked the manuscript. The changes are highlighted yellow and we have addressed your comments point by point below.

We would like to thank you once again for your constructive criticism and would appreciate your positive feedback.

We look forward to hearing from you at your earliest convenience.

Sincerely,

Prof. Dr. Irina Blumenstein

---

## [Editor Report · Decision Letter 1]

6 Oct 2025

Sequencing Therapy for Optimal Response in Mirikizumab (STORM)-Study: A Tertiary Referral Center Study on Patients with Therapy-Refractory Ulcerative Colitis

PONE-D-25-43407R1

Dear Dr. Blumenstein,

We’re pleased to inform you that your manuscript has been judged scientifically suitable for publication and will be formally accepted for publication once it meets all outstanding technical requirements.

Kind regards,

Vipula Rasanga Bataduwaarachchi, MD

Academic Editor

PLOS ONE

---

## [Editor Report · Acceptance letter]

PONE-D-25-43407R1

PLOS ONE

Dear Dr. Blumenstein,

I'm pleased to inform you that your manuscript has been deemed suitable for publication in PLOS ONE. Congratulations! Your manuscript is now being handed over to our production team.

Kind regards,

on behalf of

Dr. Vipula Rasanga Bataduwaarachchi

Academic Editor

PLOS ONE